# Comparative transcriptome analysis uncovers different heat stress responses in heat-resistant and heat-sensitive jujube cultivars

**Juan Jin**[1☯], **Lei Yang**[1☯], **Dingyu Fan**[1], **Xuxin Liu**[2], **Qing Hao**[1]*

**1** Institute of Horticultural crops, Xinjiang Academy of Agricultural Sciences, Urumqi, China, **2** Xinjiang Agricultural Vocational Technical College, Changji, China

☯ These authors contributed equally to this work.
* haoqingxj@sohu.com

**Data Availability Statement:** The RNA-seq data were deposited at the NCBI Gene Expression Omnibus (GEO) database under accession numbers GSE136383 and GSE136047.

## Abstract

Jujube (*Ziziphus jujuba* Mill.) is an economically and agriculturally significant fruit crop and is widely cultivated throughout the world. Heat stress has recently become a primary abiotic stressor limiting the productivity and growth of jujube, as well as other crops. There are few studies, however, that have performed transcriptome profiling of jujube when it is exposed to heat stress. In this study, we observed the physiochemical changes and analyzed gene expression profiles in resistant jujube cultivar 'HR' and sensitive cultivar 'HS' subjected to heat stress for 0, 1, 3, and 5d. Twenty-four cDNA libraries from 'HR' and 'HS' leaves were built with a transcriptome assay. A total of 6887 and 5077 differentially expressed genes were identified in 'HR' and 'HS' after 1d, 3d, and 5d of heat stress compared with the control treatment, GO and KEGG enrichment analysis revealed that some of the genes were highly enriched in oxidation-reduction process, response to stress, response to water deprivation, response to heat, carbon metabolism, protein processing in endoplasmic reticulum, and plant hormone signal transduction and may play vital roles in the heat stress response in jujube plants. Differentially expressed genes were identified in the two cultivars, including heat shock proteins, transcriptional factors, and ubiquitin-protein ligase genes. And the expression pattern of nine genes was also validated by qRT-PCR. These results will provide useful information for elucidating the molecular mechanism underlying heat stress in different jujube cultivars.

## Introduction

Plants have to deal with a broad array of environmental conditions, such as drought, salt, low temperatures, high temperatures, nutrition deficiency, and heavy metals, which negatively affect the growth and development of plants [1,2]. These abiotic stressors induce a number of different responses in plants, including biochemical, physiological, molecular, anatomical, and morphological changes [3,4]. Because of the increasing temperatures worldwide, heat stress has become one of the primary environmental factors restricting the distribution and

**Funding:** This research was supported financially by the National Natural Science Foundation of China (Grant # 31801815).

**Competing interests:** The authors have declare that they have no competing interests.

cultivation of crops globally, thus leading to significant reductions in yield, growth, and mortality [5]. Hence, it is an effective way to seek for the suitable heat-resistant cultivar in nature and explore its mechanism of plant stress resistance.

Plants have evolved multiple strategies to combat high temperatures in an effort to reduce the negative effects of heat stress on crop growth and development [6]. At the cellular, physiological, and cellular levels, the generation of phytohormones, antioxidants and osmoprotectants, as well as dynamic membrane regulation, and stomatal closure are associated with plant adaptive responses against heat stress [7,8]. Heat stress changes the way genes are involved in signaling pathways, as well as transcriptional control and the expression heat shock proteins at the molecular level [9]. New research has found that levels of gene expression induced by heat stress are regulated by transcriptional networks as well as the transcription factors (TFs) in plants after translational regulation [10]. For example, heat shock proteins (HSPs) are known as target genes for TFs responding to heat stress [11,12]. Many HSPs function as molecular chaperones [13] and serve important roles in the regulation of the protein quality by renaturing various proteins that have been denatured by heat stress [14]. In peanuts, accumulation of small HSPs could improve its resistance to heat [15].

In recent years, transcriptome sequencing has become an important technique for identifying stress-related genes and finding the multiple biological pathways [16]. It has been successfully used to reveal mechanisms related to abiotic stress responses in many species, such as Chinese kale (*Brassica alboglabra*) [17], *Pyropia haitanensis* [18], Chinese cabbage (*Brassica rapa* ssp. Chinensis) [19], sweet maize (*Zea mays* L.) [20], spinach (*Spinacia oleracea* L.) [21], pepper (*Capsicum annum* L.) [22]. Jujube (*Ziziphus jujuba* Mill.) is a traditional fruit crop, native to China, which belongs to the Rhamnaceae family and widely grown in Asian tropical and subtropical areas [23]. Approximately 3.25 million hectares are cultivated in China [24]. High temperature caused serious loss of quality and production during the jujube growth especially in the open field cultivation. However, little knowledge about the molecular studies on jujube under heat stress is uncovered.

In this study, we obtained two differently heat-resistant jujube cultivars and carried out RNA-sequencing analysis to explore the transcriptional changes between the two jujube cultivars under heat stress. Our objective was to identify differentially expressed genes involved in heat stress on jujube, and potential heat stress-related genes including heat shock proteins, transcriptional factors, and ubiquitin-protein ligase genes were detected. This study can help in better understanding of transcriptomic defense mechanisms associated with heat tolerance in jujube.

## Materials and methods

### Plant materials, heat stress treatments and sample collection

Ziziphus jujuba cv.'HR' and Ziziphus jujuba cv.'HS' were grown in the Forestry Management Station of Turpan city, Forestry and Grassland Administration of Turpan city, located in Turpan, Xinjiang province, China. 'HR' has been proven to be resistant to heat stress while 'HS' is susceptible. Current-growth branches of primary shoots, approximately 0.6 cm in diameter, were collected from the jujube cultivars in the end of April, 2018. Then the green stem cuttings of 'HR' and 'HS' were rooted in pots with a mixture of 1 perlite: 1 vermiculite: 1 river sand (v/v/v) and grown in a greenhouse under mist conditions. When the cuttings were rooted, they were repotted into larger pots, and grown in a greenhouse at 70–80% relative humidity. The temperature was almost kept at 30°C and the maximum light intensity was about 800 μmol m$^{-2}$s$^{-1}$ during the daytime, the temperature was maintained at 20°C during the night. About 6 weeks later, young seedlings of 'HR' and 'HS' were transferred to a controlled environment

room (14h/10h light/dark, 30/20°C day/night cycle, 70–80% relative humidity, and light intensity at 600 μmol m$^{-2}$s$^{-1}$). There were a total of 180 plants for 'HR' and 'HS' seedlings, respectively. Seedlings with eight true leaves of 'HR' and 'HS' were then treated at 45°C in the same controlled environment room (except for temperature, the other conditions were the same as above) from 8:00 am to 22:00 pm for 0d (control), 1d, 3d, 5d. Each treatment included three biological replications for each cultivar, and each biological replicate contained 15 plants. Plant samples were collected after the treatments started.

## Physiological measurements and ultra-structural observation

Before the transcriptome study, three physiological indexes, including electrolyte leakage, malondialdehyde (MDA), and proline were measured 3 times (n = 3) with 5 plants per replicate in each treatment time point for each cultivar in order to analyze the physiological changes of two jujube cultivars subjected to heat stress. Relative electrolyte leakage (relative electrolyte conductivity; REC) was used to assess the cell membrane permeability as described by Yang et al. [25]. Lipid peroxidation was determined by measuring the amount of malondialdehyde (MDA) content according to the methods of Heath [26]. Proline content was determined using the sulfosalicylicaid method [27].

For ultra-structural observation, the sixth leaf from the top was removed from 4 random plants in each treatment time point (n = 3) for each cultivar and then the leaf tissue adjacent to the vein were cut into 1 mm × 2 mm and fixed immediately by immersion in 4% glutaraldehyde overnight, then washed with 0.1 M PBS (pH7.4) for 15 min three times. After post fixation in 1% osmium tetroxide for 5 h at 20°C, the fragments were washed with 0.1 M PBS for 15 min three times and gradually dehydrated in ethanol (30, 50, 70, 80, 90,95 and 100% ethanol for 1 h), then immersed in 3 ethanol: 1acetone (v/v), 1 ethanol: 1acetone (v/v), 1 ethanol: 3 acetone (v/v) for 30 min, respectively. The fixed samples were embedded in SPIpon812 epoxy matrix (SPI, Chem) and observed by transmission electron microscopy (HT7700, Hitachi, Tokyo, Japan).

## RNA extraction, cDNA library construction and Illumina sequencing

For RNA extraction, the sixth leaf from the top was collected from 6 random plants in each treatment time point (n = 3) for each cultivar, leaf sapmles were then immediately frozen in liquid nitrogen and stored at -80°C prior to RNA extraction. Total RNA was extracted from these samples using the RNAprep Pure Plant Kit (Tiangen, Beijing, China) following the manufacturer's instructions. The purity of the extracted RNA was assessed by NanoDrop 1000 spectrophotometer (Thermo Fisher Scientific, Wilmington, DE, USA). RNA concentration was performed by using Qubit® 2.0 Flurometer (Life Technologies, Carlsbad, CA, USA). RNA integrity was checked by the RNA Nano 6000 Assay Kit of the Agilent Bioanalyzer 2100 system (Agilent Technologies, Santa Clara, CA, USA).

Construction of RNA sequencing libraries were performed at Biomarker Technologies Corporation (Beijing, China) following instructions similar to Guo et al. [17]. Briefly, after enrichment and purification with oligo (dT)-rich magnetic beads, mRNA was cleaved into short fragments. Then first- and second-strand cDNA were synthesized using the mRNA fragments as templates. The cDNA was purified by AMPure XP beads and resolved with elution buffer for end reparation and single nucleotide adenine addition, and the short fragments were connected with adapters. Then the suitable fragments were selected as templates for PCR amplification. Finally, the twenty-four cDNA libraries were sequenced using an Illumina HiSeq™ 2500. The RNA-seq data were deposited at the NCBI Gene Expression Omnibus (GEO) database under accession numbers GSE136383 and GSE136047.

### RNA-seq reads mapping and transcript assembly

After trimming adapter sequences, removing low quality reads and unknown nucleotides larger than 5%, clean reads were filtered from the raw reads. Cleaned RNA-seq reads were then mapped to the jujube reference genome retrieved from the NCBI Database (http://www.ncbi.nlm.nih.gov/genome/) [28] using Bowtie 2 (http://bowtie-bio.sourceforge.net/index.shtml) [29] and TopHat 2 (http://ccb.jhu.edu/software/tophat/index.shtml) [30]. Then the Sequence Alignment Map (http://samtools.sourceforge.net) [31] files were generated by TopHat2 and subsequently transcripts were assembled by Cufflinks (http://cufflinks.cbcb.umd.edu/) [32].

### Differentially expressed genes and functional analysis

Differentially expressed genes (DEGs) of different libraries were analyzed using the FPKM (the fragments per kilobase of exon per million fragments mapped reads) method and the Bioconductor software package edgeR was used to screen out the DEGs [33]. Genes were determined to be differentially expressed based on the fold change (FC $\geq$ 2 or $\leq$ 0.5) and a false discovery rate (FDR $\leq$ 0.01).

Gene ontology (GO) and Kyoto Encyclopedia of Gene and Genomes (KEGG) enrichment analysis of the DEGs were performed using the GOseq R packages [34] and KOBAS software [35], respectively.

### Validation of RNA-seq by quantitative real-time PCR (qRT-PCR)

The qRT-PCR was performed on RNA extracted from leaf samples of both cultivars at the four heat treatment time points as described by Bu et al. [36] using the *ZjActin* as the internal control to normalize the expression data. Primers used for qRT-PCR were listed in **S1 Table**. We used the same samples of RNA for both the qRT-PCR verification and the RNA-seq. 1 μg of RNA was reversely transcribed using OneScript® Two-Step RT-PCR Kit (Applied Biological Materials (ABM), Vancouver, Canada) according to the instructions of the manufacturer. We performed the experiment on the LightCycler96 with BrightGreen Express 2X qPCR Master-Mix (ABM, Canada), according to the following PCR program: 2 min of denaturing at 95°C, 5 s of denaturing at 95°C, performed 40 times, 30 s of extension and annealing at 60°C. Gene expression was calculated by the $2^{-\Delta\Delta Ct}$ method [37]. All qRT-PCR were repeated with three technical and three biological replicates.

### Statistical analysis

One-way analysis of variance (ANOVA) was conducted using SPSS 17.0 statistical software (SPSS, IL, USA). We used Duncan's Multiple Range Test to analyze the differences among means, and considered results to be significant at a p-value $\leq$ 0.05.

## Results

### Physiochemical changes of two jujube cultivars in response to heat stress

To describe the heat tolerance phenotype of the heat-resistant 'HR' and heat-sensitive 'HS' jujube cultivars, young jujube seedlings with uniform growth during the eight-leaf stage were exposed to 45°C of heat stress for 0, 1, 3, and 5 d, respectively. Our ultrastructure observation indicated that (a) the nucleoli of heat-resistant jujube cultivar 'HR' and heat-sensitive jujube cultivar 'HS' were clearly visible, chloroplasts were long spindles that regularly grew near the cell wall and contained white starch grains in the control treatment (0 d); (b) the nucleoli of two jujube cultivars gradually disappeared as the heat stress time extended; (c) chloroplasts of

'HR' could basically maintain a relatively normal shape, and the starch grains were faintly visible as the heat stress time prolonged; however, the chloroplasts of 'HS' started to swell and became round, detached from the cell wall and floated into the cell, and the starch granules gradually disappeared; (d) osmiophilic granules of 'HS' were significantly more than that of 'HR' after heat stress at 3, 5d (Fig 1A).

In order to further confirm the difference in heat tolerance between the heat-resistant jujube cultivar 'HR' and the heat-sensitive jujube cultivar 'HS', we measured the electrolyte leakage, MDA and proline content. Compared with the control treatment (0 d), there was no significant difference in the relative electrolyte leakage and MDA content in 'HR' leaves after 1, 3 d of heat stress, while exposure to heat stress for 5 days significantly increased the relative electrolyte leakage and MDA content by 18.04% and 36.14%, respectively (Fig 1B and 1C). For heat-sensitive jujube cultivar 'HS', exposure to heat stress for 1, 3 days significantly increased the relative electrolyte leakage by 25.36%, 37.97%, respectively, and the MDA content were significantly increased by 34.93%, 75.69% and 117.29% after 1, 3,5 d of heat stress, respectively.

Compared with the control treatment (0 d), exposure to heat stress for 3, 5 days significantly increased the proline content in the leaves of 'HR' and 'HS', among which the proline

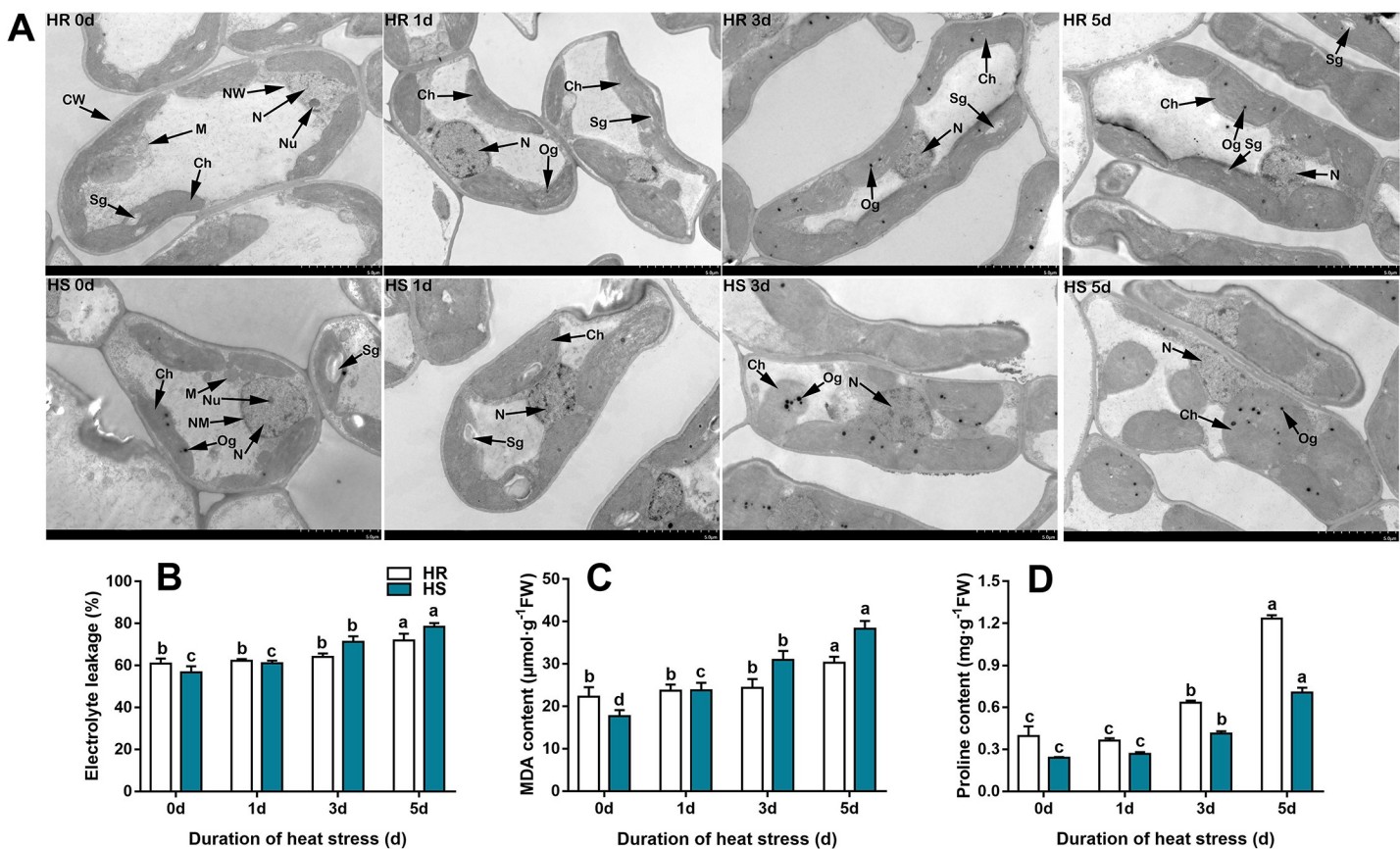

**Fig 1. Ultrastructure observation and physiological analyses of heat-resistant 'HR' and heat-sensitive 'HS' jujube cultivars subjected to heat stress.** (A) Characteristic structures in non-stressed (0 d) and heat stressed leaves observed by transmission electron microscopy. CW cell membrane, NM nucleus membrane, N nucleus, Nu nucleolus, Ch chloroplast, M mitochondrion, Sg starch grain, Og osmiophilic granule. The magnification of these pictures is × 2.5k. The bars represent 5 μm. (B) Changes in electrolyte leakage, (C) MDA content, and (D) proline content in leaves of two jujube cultivars exposed to 45˚C for 0, 1, 3, and 5 d, respectively. Error bars are ± SD of the mean (n = 3). Samples were collected from nine plants for each replicate. Duncan's multiple range test (*P* < 0.05) was used to determine significant differences between treatments. These differences are shown by the letters above the bars.

content of 'HR' were increased by 59.53%, 211.75%, respectively, and the proline content of 'HS' were increased by 54.83%, 165.09%, respectively. No significant differences were observed in the proline content of the two jujube cultivars between the 1d of heat stress and the control treatments (**Fig 1D**). In addition, the proline content of 'HR' was always higher than that of 'HS' among all the experimental treatments. These results indicated that 'HS' was sensitive while 'HR' was tolerant to heat stress.

### Illumina sequencing and alignment to the reference genome

In order to determine the transcriptome profiles of 'HR' and 'HS' in response to heat stress, we performed RNA-Seq analyses on 'HR' and 'HS' at 0, 1, 3, and 5 d. Three biological replicates were made at each time point for both cultivars. We established 24 RNASeq cDNA libraries (HR-0-a, HR-0-b, HR-0-c; HR-1-a, HR-1-b, HR-1-c; HR-3-a, HR-3-b, HR-3-c; HR-5-a, HR-5-b, HR-5-c; HS-0-a, HS-0-b, HS-0-c; HS-1-a, HS-1-b, HS-1-c; HS-3-a, HS-3-b, HS-3-c; HS-5-a, HS-5-b, HS-5-c). After filtering and trimming, 22,595,993–32,752,424 clean reads (the percentage of Q30 and GC being 91.25–94.79% and 44–45.97%) were obtained (**S2 Table**), Most reads (81.91–87.76%) could be mapped to the jujube reference genome, among which 71.08–76.71% were uniquely mapped ones. Box-plots displayed the range of the fragments per kilobase of exon per million fragments mapped (FPKM) values in all the 24 libraries (**Fig 2A**). Sample clustering analysis presented 4 Clusters. We found that Cluster 1 only gathered samples of 'HR', Cluster 2 assembled mostly samples of 'HS', Cluster 3 and 4 grouped about the

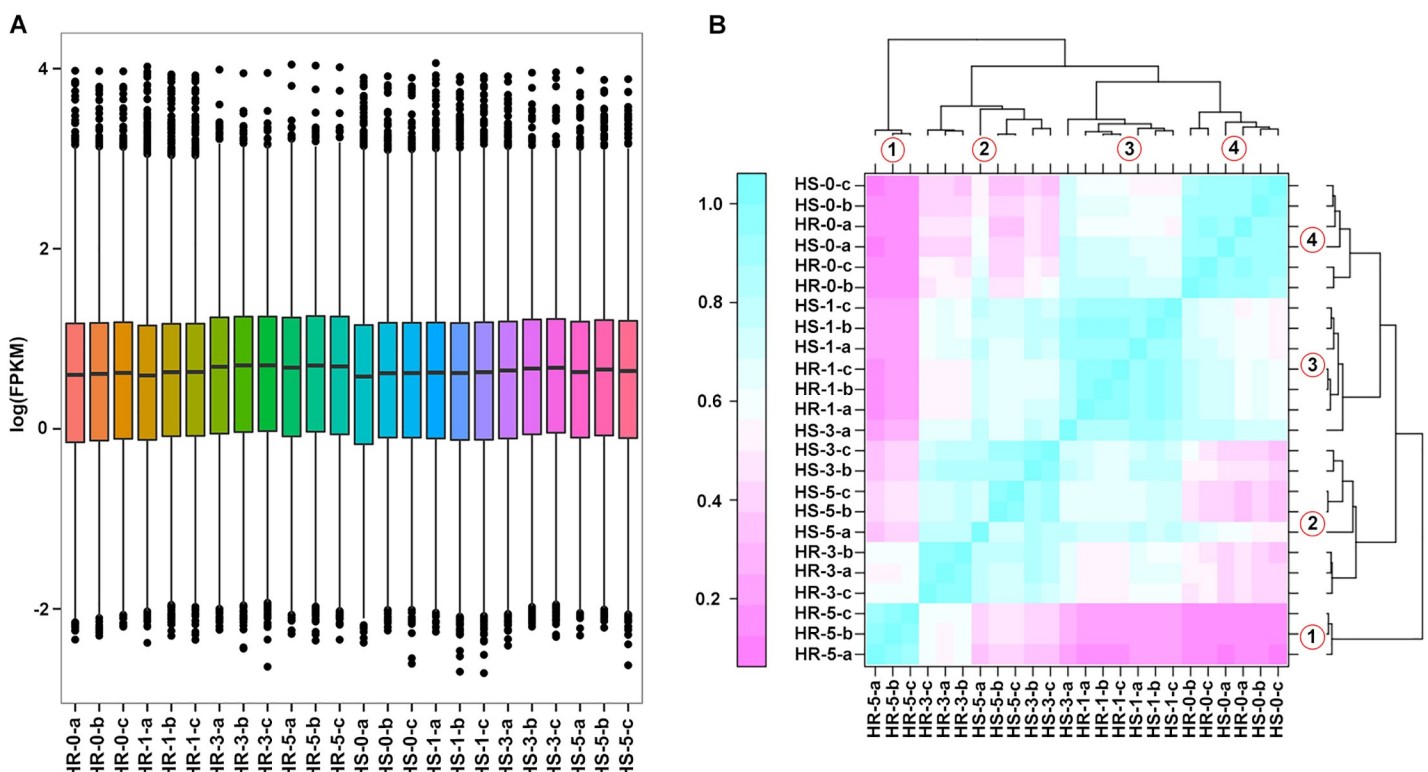

**Fig 2. Overview of the Illumina transcriptome sequencing.** (A) Gene expression levels of all the 24 libraries. HR represents the heat resistant jujube cultivar, HS represents the heat sensitive jujube cultivar. (B) Heatmap clustering showing the sample correlation analysis of the 24 sequenced samples. The number 1, 2, 3 and 4 represent the 4 Clusters of samples.

same number of samples in both 'HR' and 'HS', these results indicated that the gene expression profile in 'HR' is quite different from that in 'HS' after 5 d of heat stress (**Fig 2B**).

## Explorations of differentially expressed genes between 'HR' and 'HS' cultivars

To identify the number of DEGs between 'HR' and 'HS' after 0, 1, 3, 5d of heat stress, we analyzed the gene expression patterns using the DEGseq [33] based on the Fold Change ≥2 and FDR <0.01.

In total, 6887 and 5077 differentially expressed genes were identified in 'HR' and 'HS', respectively (**S3 Table**). As shown in Table 1, 1880, 4350 and 5435 annotated genes were differentially expressed in 'HR' after heat stress for 1, 3 and 5d, respectively; among which, 1078 up-regulated genes and 802 down-regulated genes were identified in the HR1d vs HR0d comparison; in the HR3d vs HR0d comparison, the number of up-regulated genes and down-regulated genes was 2077 and 2273, respectively; in the HR5d vs HR0d comparison, the numbers were 2315 and 3120, respectively (**S4 Table**).Meanwhile, There were 2390, 2998, and 4064 annotated genes differentially expressed in 'HS' after heat stress for 1, 3 and 5d, respectively; in the HS1d vs HS0d comparison, the number of up-regulated genes and down-regulated genes was 1035 and 1355, respectively; in the HS3d vs HS0d comparison, 1327 genes were up-regulated and 1671 genes were down-regulated; in the HS5d vs HS0d comparison, there were 1759 up-regulated genes and 2305 down-regulated genes.

Venn diagram analysis showed that 487 up-regulated and 547 down-regulated genes overlapped in the HR1d vs HR0d, HR3d vs HR0d, HR5d vs HR0d comparisons, whereas 555 common genes were found to be up-regulated and 951 down-regulated in the HS1d vs HS0d, HS3d vs HS0d, HS5d vs HS0d comparisons (**Fig 3**).

Furthermore, we identified the genes expressed under the same heat stress time period between 'HR' and 'HS' (Table 1). In the control treatment (0 d), 179 genes were up-regulated and 158 genes were down-regulated in the HR0d vs HS0d comparison; After being subjected to heat stress, 203 genes were up-regulated and 402 genes were down-regulated in the HR1d vs HS1d comparison; 388 genes were up-regulated and 444 genes were down-regulated in the HR3d vs HS3d comparison; and 860 genes were up-regulated and 543 genes were down-regulated in the HR5d vs HS5d comparison. Venn diagram results showed that 5 genes (LOC107406551, LOC107434446, LOC107412389, Ziziphus_jujuba_newGene_4043, Ziziphus_jujuba_newGene_7074) were down-regulated in the four comparisons (**Fig 4**).

**Table 1. The number of DEGs in different comparisons.**

| DEGs | All DEGs | | Up-regulated | Down-regulated |
|---|---|---|---|---|
| | Annotated | New discovered | | |
| HR1d_vs_HR0d | 1880 | 56 | 1078 | 802 |
| HR3d_vs_HR0d | 4350 | 95 | 2077 | 2273 |
| HR5d_vs_HR0d | 5435 | 102 | 2315 | 3120 |
| HS1d_vs_HS0d | 2390 | 68 | 1035 | 1355 |
| HS3d_vs_HS0d | 2998 | 85 | 1327 | 1671 |
| HS5d_vs_HS0d | 4064 | 86 | 1759 | 2305 |
| HR0d_vs_HS0d | 337 | 18 | 179 | 158 |
| HR1d_vs_HS1d | 605 | 23 | 203 | 402 |
| HR3d_vs_HS3d | 832 | 39 | 388 | 444 |
| HR5d_vs_HS5d | 1403 | 37 | 860 | 543 |

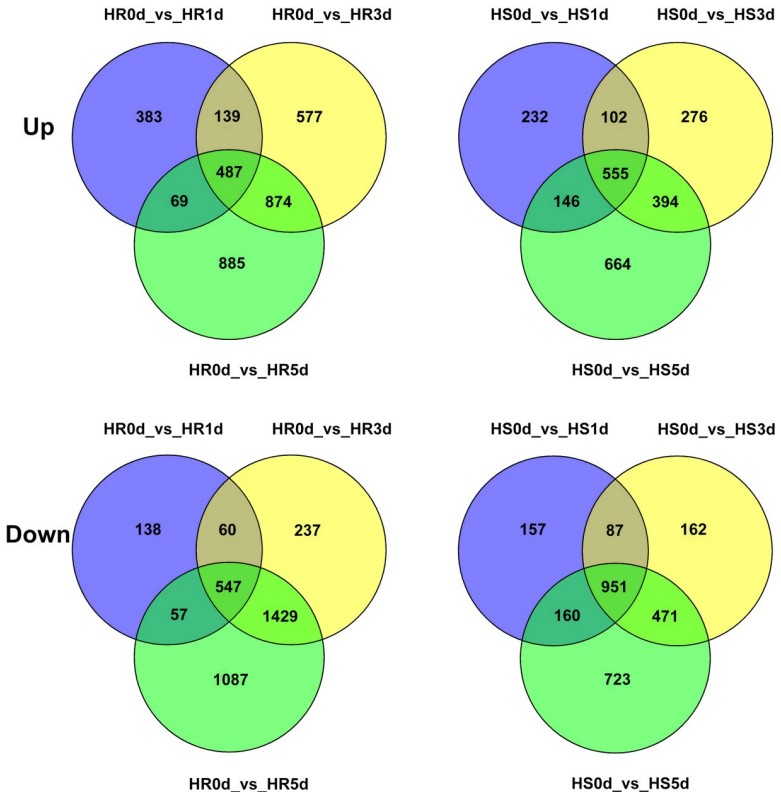

**Fig 3. Venn diagrams of the overlapped DEGs identified in 'HR' and 'HS' under heat stress.**

## Functional analysis of DEGs in 'HR' and 'HS' cultivar

To functionally annotate the DEGs in 'HR' and 'HS' jujube cultivar after different time of heat stress (1d vs.0d, 3d vs.0d, and 5d vs.0d), we aligned the DEGs against the Gene Ontology (GO)

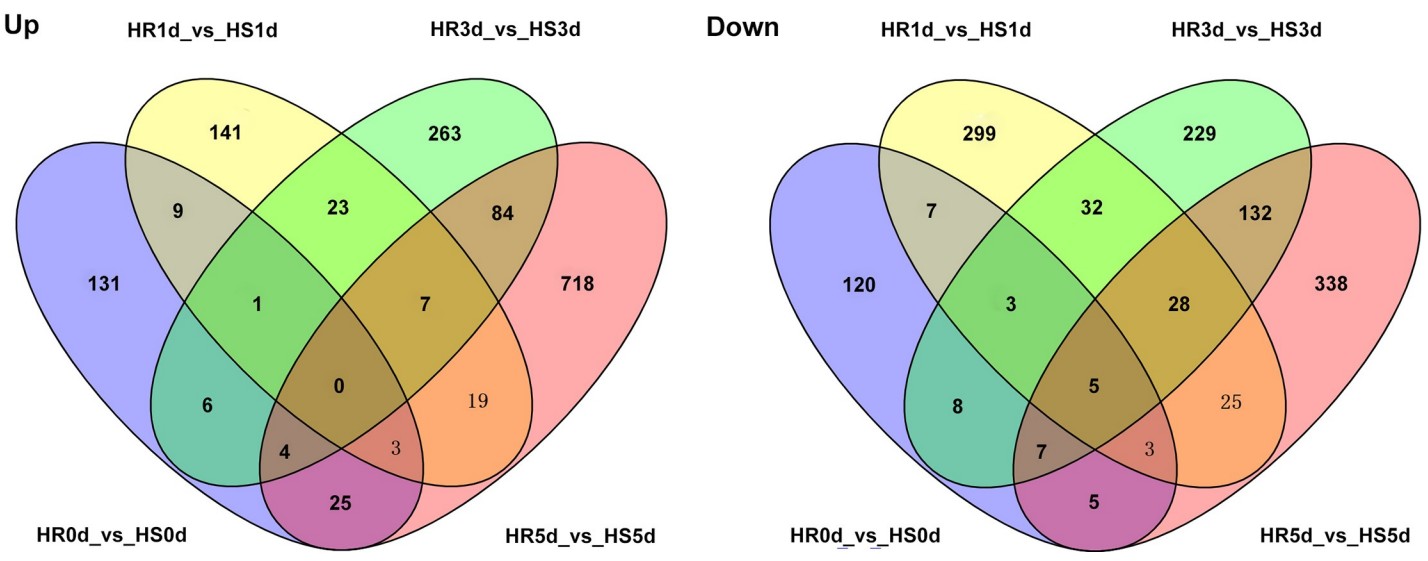

**Fig 4. Venn diagrams of up- and down-regulated genes of different heat stress time points between 'HR' and 'HS' after 0, 1, 3, 5d of heat stress.**

and Kyoto and Encyclopedia of Genes and Genomes (KEGG) database. GO enrichment analysis showed 61, 101, 116 GO terms were identified in HR1d vs. HR0d, HR3d vs. HR0d, HR5d vs. HR0d, respectively ($P$< 0.05) (**S5 Table**). There were 99, 107, 109 GO terms identified in HS1d vs. HS0d, HS3d vs. HS0d, HS5d vs. HS0d, respectively. Multiple DEGs were enriched in "oxidation-reduction process" (GO:0055114), "response to stress" (GO:0006950), "response to water deprivation" (GO:0009414), and "response to heat" (GO:0009408) terms among all the six comparisons (**Fig 5**, **S5 Table**), indicating that genes involved in these biological processes may participate in responding to heat stress.

KEGG analysis was performed to identify the potential biological pathways of genes represented in the transcriptome after heat stress in jujube. Results showed that 15, 27, and 27 pathways were significantly enriched for DEGs in HR1d vs. HR0d, HR3d vs. HR0d, HR5d vs. HR0d, respectively, and 32, 28, 27 pathways were significantly enriched in HS1d vs. HS0d, HS3d vs. HS0d, HS5d vs. HS0d, respectively (**S6 Table**). The pathways enriched for DEGs of HR1d vs. HR0d were "carbon metabolism" (ko01200), "protein processing in endoplasmic reticulum" (ko04141), and "biosynthesis of amino acids" (ko01230) (**Fig 6A**); the pathways enriched for HS1d vs. HS0d DEGs were "carbon metabolism" (ko01200) and "biosynthesis of amino acids" (ko01230) (**Fig 6B**), indicating that genes in these pathways may be involved in early heat stress response. The pathways enriched for HR3d vs. HR0d and HS3d vs. HS0d DEGs were "carbon metabolism" (ko01200), "biosynthesis of amino acids" (ko01230), and "protein processing in endoplasmic reticulum" (ko04141) (**Fig 6C and 6D**). The pathways enriched for HR5d vs. HR0d DEGs were "carbon metabolism" (ko01200), "starch and sucrose metabolism" (ko00500), "plant hormone signal transduction" (ko04075), "glycolysis / gluconeogenesis" (ko00010) (**Fig 6E**); the pathways enriched for HS5d vs. HS0d DEGs were "carbon metabolism" (ko01200), "glycolysis / gluconeogenesis" (ko00010) and "glycine, serine and threonine metabolism" (ko00260) (**Fig 6F**), suggesting that these metabolism processes possibly play important roles in response to heat after 5 days. Additionally, heat shock protein genes belonging to the protein processing in endoplasmic reticulum (ko04141) pathway were significantly enriched in HR1d vs. HR0d, suggesting that 'HR' can sharply accumulate heat shock proteins to improve the heat tolerance of the plants when suffering from heat stress.

## HSPs related to heat stress

HSPs are well known to play important roles in responding to heat stress. In this study, four *HSP90s*, two *HSP70s*, and eighteen *sHSPs* were identified as candidate genes in different HSP families, among which, *sHSPs* including *HSP26.5*, *HSP22.0*, *HSP18.5*, *HSP17.4*, *HSP17.1* and *HSP16.9*. All *HSPs* (except LOC107413527) were significantly up-regulated in both 'HR' and 'HS' jujube cultivar under heat stress (**Fig 7**). In addition, one DEG (LOC107420074) was up-regulated only in 'HR' and one DEG (LOC107429130) was only up-regulated in 'HS'. Two *sHSPs* (LOC107407995, LOC107410965) were sharply responsive to heat stress in 'HR' at 1 d and the expression levels increased after 3, 5 d of heat stress, while the two *sHSPs* were identified to be up-regulated in 'HS' only after 5 d of heat stress.

## Transcription factors involved in heat stress

In our present study, 86 and 52 DEGs were respectively identified to be involved in eight TF families in 'HR' and 'HS' during the heat stress (**S7 Table**). These TF families include *ERF*, *WRKY*, *MYB*, *NAC*, *DERB*, *bHLH*, *HSF* and *C2H2* (**Table 2**). There were 21 common transcription factors in HR1d vs.HR0d, HR3d vs.HR0d, and HR5d vs.HR0d, among which, 16 genes encoding four *ERFs* (LOC107432807, LOC107404986, LOC107408332, LOC107417413), six *NACs* (LOC107419222, LOC107421097, LOC107428947,

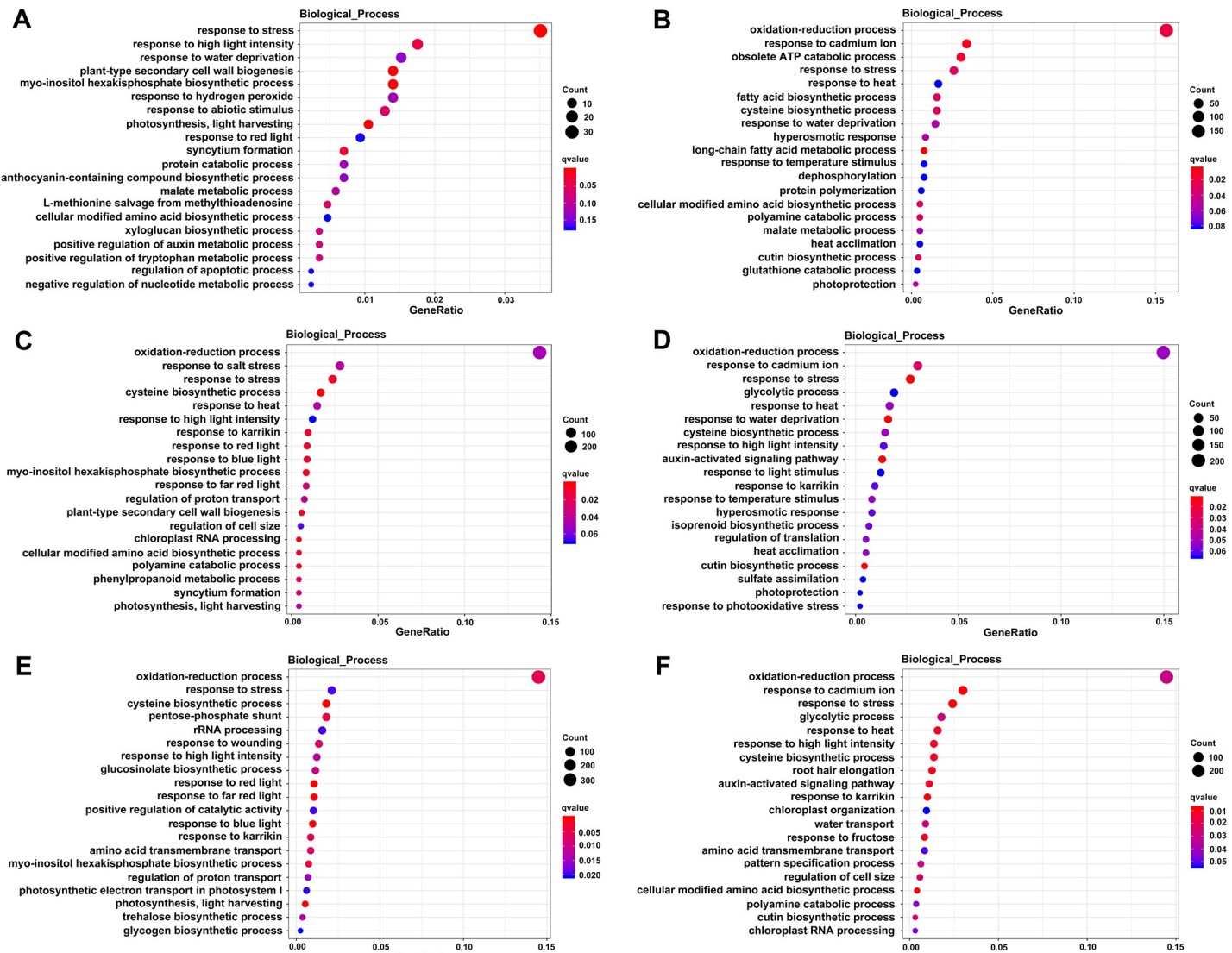

**Fig 5. Scatterplot of enriched GO pathways between the two cultivars after heat stress.** (A), (C) and (E) represent enriched GO pathways of the DEGs in HR 1d vs. HR0d, HR 3d vs. HR0d, and HR 5d vs. HR0d, respectively; (B), (D) and (E) represent enriched GO pathways of the DEGs in HS 1d vs. HS0d, HS 3d vs. HS0d, and HS 5d vs. HS0d, respectively. The horizontal axes represent the enriched GO pathways, vertical axes represent the GeneRatio of each GO pathway. GeneRatio refers to the ratio of the number of DEGs enriched in certain GO pathway to the total number of differentially expressed genes. The greater the value is, the higher the number of DEGs is. The size of the dots indicates the number of DEGs enriched in certain pathway, and the color of the dots corresponds to the range of the q value (adjusted p value). Only the top 20 terms are listed here.

LOC107435293, LOC107406551, and Ziziphus_jujuba_newGene_7074), one *MYB* (LOC107426233), one *bHLH* (LOC107421731), two *zinc finger proteins* (LOC107417576, LOC107424717), two *HSFs* (LOC107431837, LOC107429964) were up-regulated and 5 genes encoding two *ERFs* (LOC107435298, LOC107415044), one *WRKY* (LOC107432867), one *bHLH* (LOC107432409), one *zinc finger protein* (LOC107416167) were down-regulated in 'HR' at all the heat stress time points. In addition, 15 common transcription factor genes encoding six *ERFs* (LOC107404986, LOC107405213, LOC107408332, LOC107424539, LOC107435298, LOC107415044), two *NACs* (LOC107406551, LOC107416163), one *MYB* (LOC107412845), two *bHLHs* (LOC107421731, LOC107432409), two *zinc finger proteins*

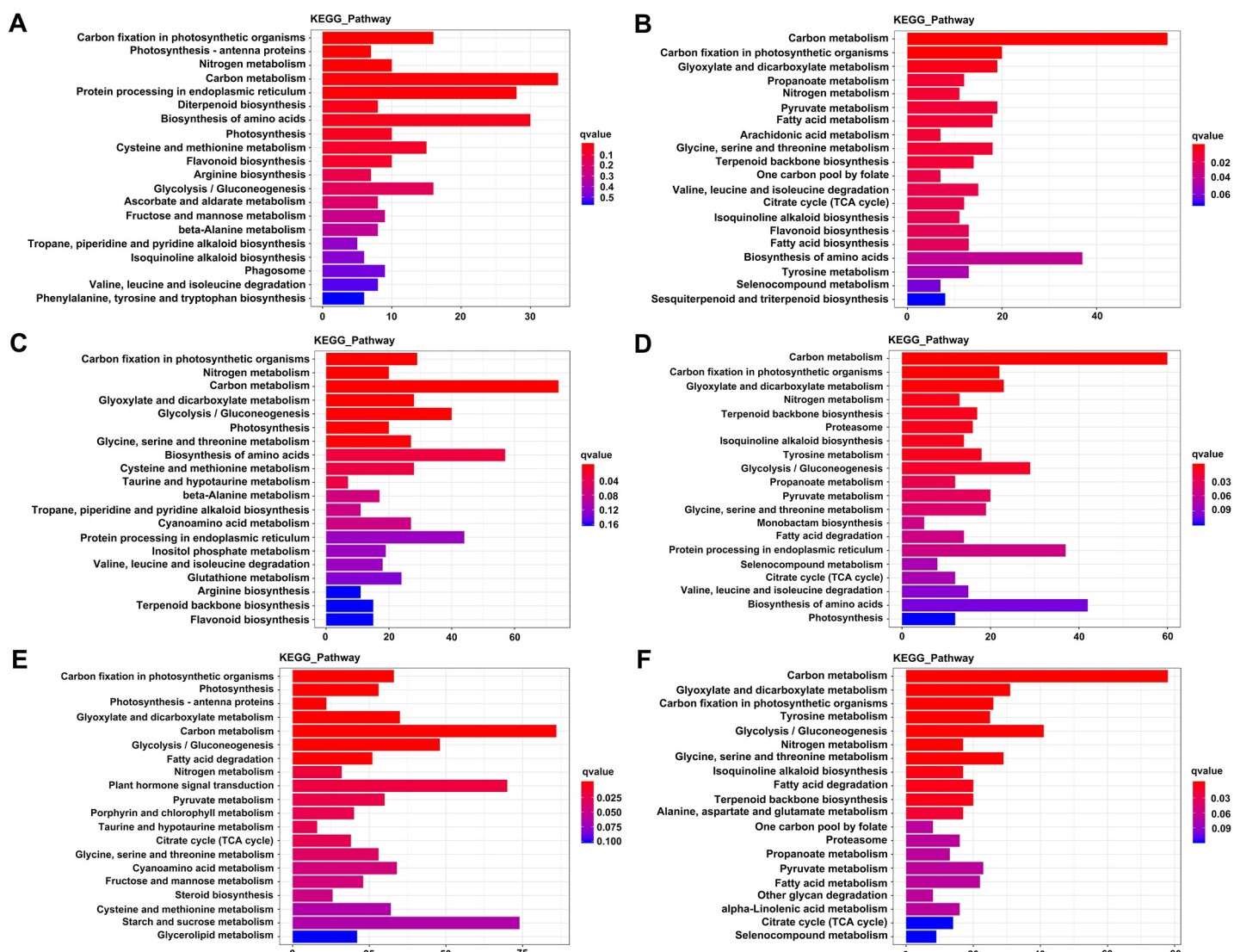

**Fig 6. Scatterplot of enriched KEGG pathways between the two cultivars after heat stress.** (A), (C) and (E) represent enriched KEGG pathways of the DEGs in HR 1d vs. HR0d, HR 3d vs. HR0d, and HR 5d vs. HR0d, respectively; (B), (D) and (E) represent enriched KEGG pathways of the DEGs in HS 1d vs. HS0d, HS 3d vs. HS0d, and HS 5d vs. HS0d, respectively. The horizontal axes represent the enriched KEGG pathways, vertical axes represent the number of DEGs enriched in each KEGG pathway. The bar indicates the number of DEGs enriched in certain pathway, and the color of the bars corresponds to the range of the q value (adjusted p value). The longer the bar is, the more the number of DEGs is. Only the top 20 terms are listed here.

(LOC107417576, LOC107424717), two *HSFs* (LOC107431837, LOC107429964) were identified in HS1d vs. HS0d, HS3d vs. HS0d, and HS5d vs. HS0d. All of the comparisons shared 11 transcription factors, including 8 up-regulated genes and 3 down-regulated genes at all the stages of heat stress in 'HR' and 'HS'. One heat stress transcription factor C1 (LOC107431837) was found to be up-regulated in both 'HR' and 'HS' during the heat treatment, however, *HSFC1* (LOC107431837) had a 2.39, 3.47, 3.54 fold changes after 1, 3, 5d of heat stress in 'HR', but 1.57, 1.89, 1.94 fold changes in 'HS', respectively, which showed an dramatic increase in expression in 'HR' than that observed in 'HS'. Furthermore, the NAC2 (LOC107421097) gene was highly expressed in 'HR' at all the heat stress time points.

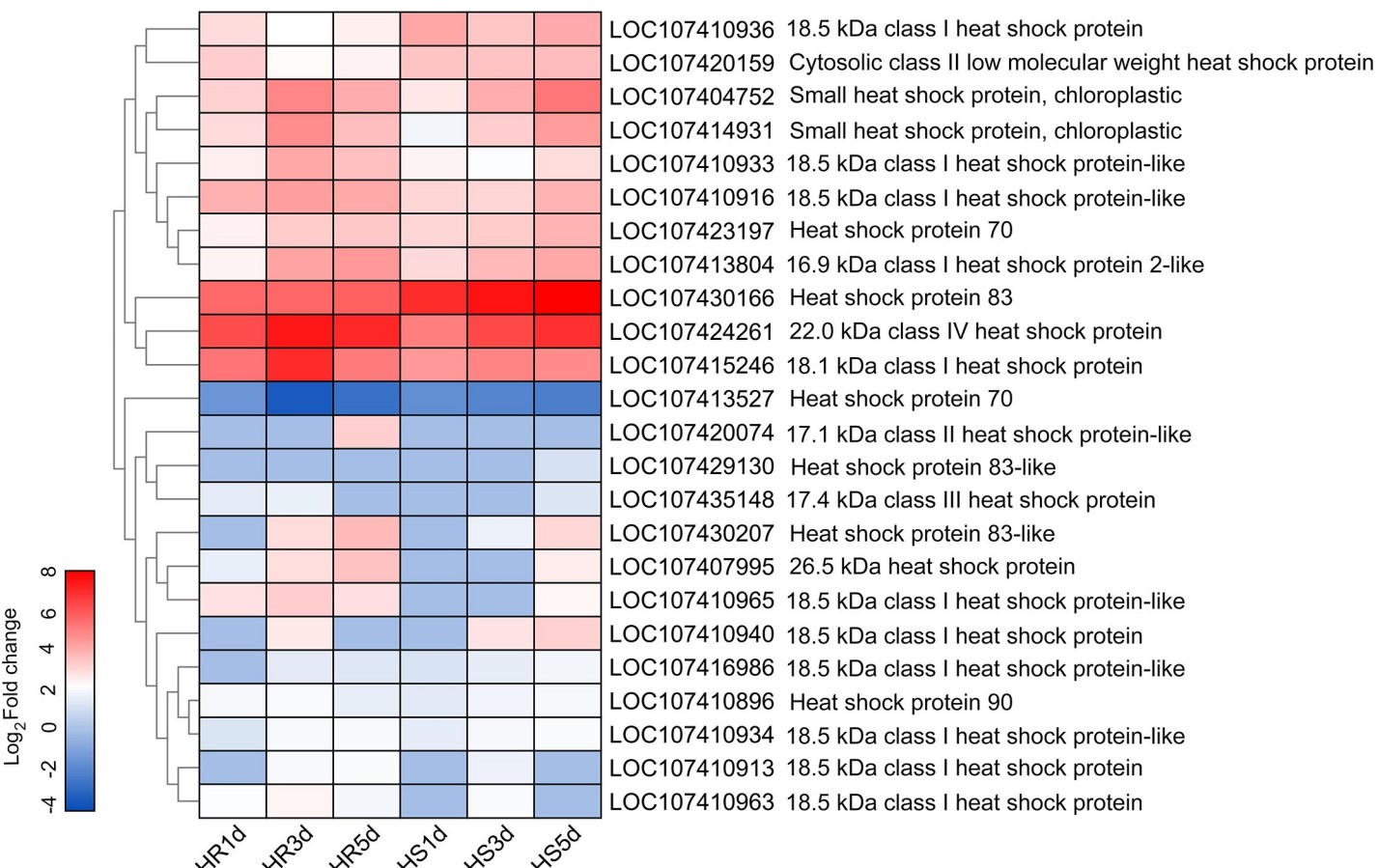

**Fig 7. Cluster analysis of DEGs identified as HSP families.** Horizontal axes represent 1, 3, and 5 d (from left to right) of heat stress in HR and HS, vertical axes represent gene ID and gene description. Heat map shows the log₂ Fold change values ranging from blue (low expression) to red (high expression).

## Expression analysis of ubiquitin-protein ligase under heat stress

In our study, we identified nineteen ubiquitin-protein ligase genes the expression patterns were shown in **Fig 8**. Four genes encoding ubiquitin-protein ligase (LOC107414605, LOC107432264, LOC107418004, LOC107428857) were all down-regulated in both 'HR' and

**Table 2. The number of DEGs identified as transcription factors in jujube leaves under heat stress.**

| Category | Heat-Resistant, HR | | | | | | Total | Heat-Sensitive, HS | | | | | | Total |
|---|---|---|---|---|---|---|---|---|---|---|---|---|---|---|
| | 1d | | 3d | | 5d | | | 1d | | 3d | | 5d | | |
| | Up | Down | Up | Down | Up | Down | | Up | Down | Up | Down | Up | Down | |
| ERF | 6 | 3 | 8 | 8 | 5 | 9 | 19 | 4 | 3 | 9 | 3 | 5 | 5 | 15 |
| WRKY | 4 | 1 | 0 | 1 | 1 | 3 | 7 | 0 | 0 | 0 | 0 | 1 | 1 | 2 |
| NAC | 7 | 0 | 10 | 0 | 9 | 0 | 10 | 6 | 0 | 4 | 0 | 7 | 1 | 10 |
| MYB | 6 | 1 | 4 | 5 | 5 | 6 | 17 | 0 | 2 | 1 | 2 | 0 | 4 | 5 |
| DREB | 0 | 0 | 0 | 1 | 1 | 1 | 2 | 0 | 0 | 0 | 1 | 0 | 1 | 1 |
| bHLH | 1 | 1 | 1 | 2 | 1 | 3 | 5 | 1 | 2 | 1 | 1 | 1 | 3 | 4 |
| C2H2 | 2 | 2 | 7 | 7 | 9 | 10 | 20 | 3 | 0 | 3 | 1 | 5 | 3 | 10 |
| HSF | 3 | 0 | 6 | 0 | 3 | 0 | 6 | 2 | 0 | 3 | 0 | 4 | 0 | 5 |

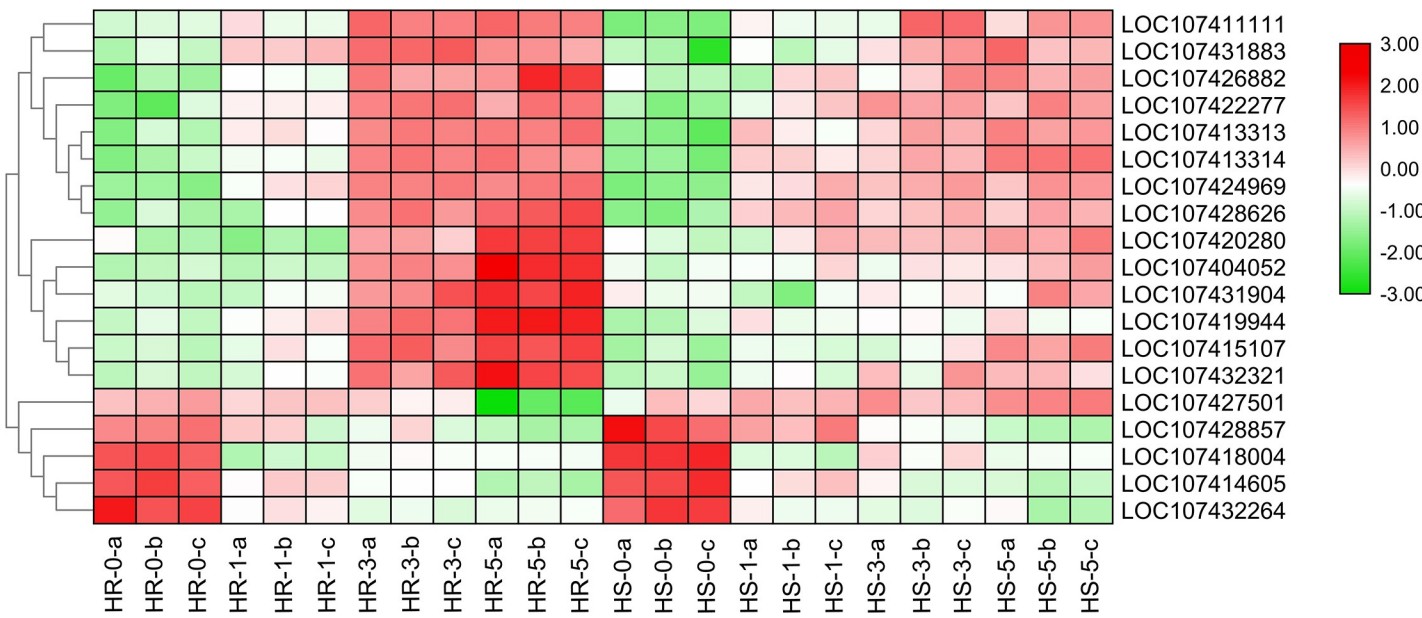

**Fig 8. Expression patterns of twenty ubiquitin-protein ligase genes in 'HR' and 'HS' after 0, 1, 3, 5d of heat stress.** Gene expression level was measured using the FPKM method. Vertical axes represent gene ID, and horizontal axes represent 0, 1, 3, and 5 d (from left to right) of heat stress in HR and HS. Heat map shows the $\log_2$ FPKM values ranging from green (low expression) to red (high expression).

'HS' under heat stress, suggesting that these four genes were repressed when jujube suffered from heat stress. In addition, the expression of fifteen genes encoding ubiquitin-protein ligase (LOC107428626, LOC107424969, LOC107413314, LOC107413313, LOC107422277, LOC107426882, LOC107420280, LOC107415107, LOC107432321, LOC107419944, LOC107404052, LOC107431904, LOC107411111, LOC107431883, LOC107403709) were all higher in 'HR' than in 'HS' at 3, 5d.

## Validation of DEGs by qRT-PCR

To confirm the validation of the RNA-seq data, nine DEGs were selected for real-time PCR analysis and they included HSP16.9 (LOC107413804), HSP70 (LOC107423197), HSP83-like (LOC107430207), HSFc-1(LOC107431837), HSP transcription factor (LOC107429964), bHLH48-like(LOC107432409), E3 ubiquitin-protein ligase RHA2A (LOC107413313), E3 ubiquitin-protein ligase SINAT2 (LOC107415107), and E3 ubiquitin-protein ligase RHF1A (LOC107426882). As shown in **Fig 9**, the expression patterns of both qRT-PCR and RNA-seq data were highly consistent.

## Discussion

Global warming is becoming a threat to agricultural production in many areas across the world [38]. Jujube (*Ziziphus jujuba* Mill.) is a traditionally popular fruit crop that is native to China [39]. However, high temperature stress significantly reduces the jujube yield in the Xinjiang Region of north China. It is urgent to elucidate the molecular mechanisms by which plants adapt to heat stress. Therefore, we explored both the physiochemical and transcriptomic changes between the resistant and sensitive cultivar in response to heat stress. Our ultrastructural observation clearly showed that the time-course heat stress exerted a more conspicuous effect on the heat-sensitive jujube cultivar 'HS' than heat-resistant jujube cultivar 'HR', the chloroplasts became more globular and the number of osmiophilic granules increased in

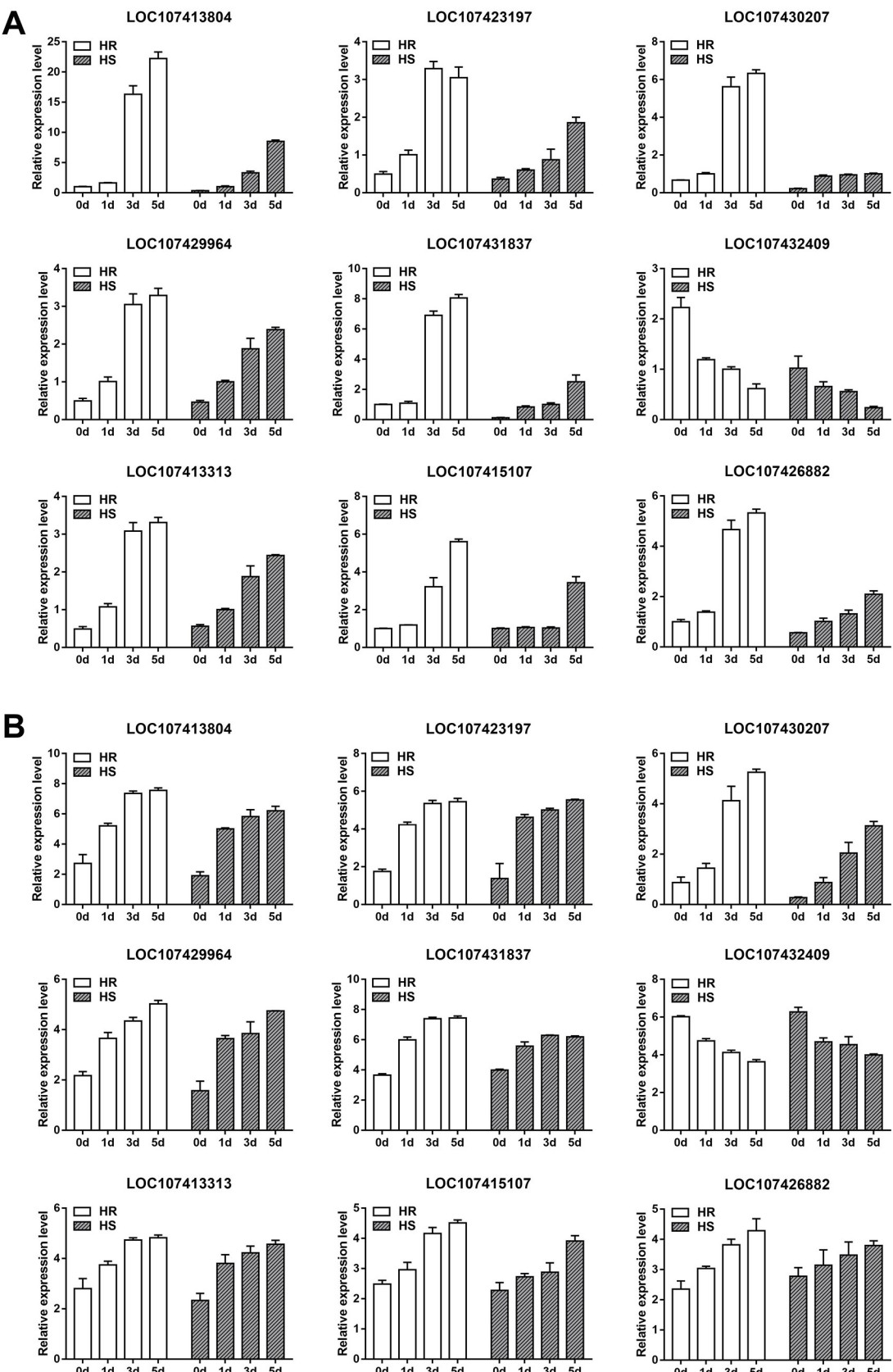

**Fig 9. Expression analysis of nine selected DEGs in 'HR' and 'HS' after 0, 1, 3, 5d of heat stress.** (A) qRT-PCR results of nine DEGs. (B) mRNA expression levels of the same DEGs determined by RNA-seq, $Log_2$(FPKM) was used to calculate the expression levels of genes. Horizontal axes represent 0, 1, 3, and 5 d (from left to right) of heat stress in HR and HS, the vertical axes represent the expression levels. The white bar presents the expression levels in 'HR', and the grey bar presents the expression levels in 'HS'. The error bars represent SD (n = 3).

response to heat, which were consistent with the leaf ultrastructure changes of Ara et al. [40] in cucurbits. The nucleolus has been considered as a sensor of cell stress since it can suffer alterations of morphology, area, and number per nucleus in response to stress [41,42]. In our study, we found that the nucleoli of two jujube cultivars were sensitive to heat stress and gradually disappeared, which had been reported in cowpea cells after exposure to elevated temperature of 45°C [43]. Moreover, we found that the starch grains of 'HR' were faintly visible, while the number of starch granules in 'HS' decreased and gradually disappeared, this phenomenon may be correlated with glucose starvation upon a decrease in photosynthetic activity under heat stress [44], confirming that 'HS' was susceptible to heat stress.

In plants, heat stress induces the accumulation of reactive oxygen species (ROS) and invokes oxidative damage [12,45]. Generating reactive oxygen species under heat stress is a symptom of cellular damage, where membrane lipids and pigments peroxidation compromise membrane permeability and function [2]. An immediate reliable indicator of heat stress is to measure the membrane integrity. In heat-tolerant genotypes, the membrane usually maintains its integrity [44]. In our study, heat stress increased the relative electrolyte leakage and MDA content in the two jujube cultivars, however, 'HS' suffered a greater degree of membrane damage than 'HR' (Fig 1). Similar results were reported by Kumar et al. [46] and Li et al. [22]. Under heat stress conditions, the accumulation of proline may act as an osmoprotectant for cellular structures in response to high temperature, which might contribute to greater heat tolerance [46]. Our results showed that higher proline content was accumulated in heat-tolerant jujube cultivar 'HR', this finding confirmed the results of study on the heat tolerant genotypes, such as maize [46] and sorghum [47], indicating that proline can be strongly correlated with the capacity of genotypes to survive heat stress.

In order to further explore the mechanisms of jujube tolerance against heat stress at the molecular level, we performed comparative RNA sequencing to reveal the differential gene expression between heat-resistant 'HR' and heat-sensitive 'HS' jujube cultivar. The high transcriptomic correlation of the three biological replicates of each sample was verified by Pearson's correlation and PCA analysis (Fig 2), which supported the reproducibility of the RNA-Seq data. Moreover, there were distinct differences in the transcriptome levels between the two cultivars in their responses to heat stress. The differential expression analysis of RNA-seq data in 'HR' and 'HS' showed that the number of DEGs at the different time points during the heat treatment of 'HR' were obviously higher than in 'HS', except at 1 d (Table 1), while the number of DEGs in 'HR' increased from 1880 at 1 d to 4350 at 3 d, these numbers only increased from 2390 to 2998 in 'HS' over the same period. These results indicated that 'HR' was better able to increase transcriptional regulation in response to high temperature stress than 'HS'. A previous study also discovered an obvious expressional divergence between heat-tolerant and sensitive *Pyropia haitanensis* strains in response to high temperature stress, with the numbers of DEGs being far greater in the heat-tolerant (THT) than in the heat-sensitive (WHT) *P. haitanensis* strains at 3 h, 6 h, and 24 h time points [18].

GO and KEGG enrichment analysis showed that some of DEGs between 'HR' and 'HS' were mainly enriched in "oxidation-reduction process", "response to stress", "response to water deprivation", and "response to heat", "carbon metabolism", "protein processing in endoplasmic reticulum", and "plant hormone signal transduction", which may reflect their similar

response to heat stress. Notably, genes involved in photosynthesis" showed significantly different expression profiles between 'HR' and 'HS', the photosynthetic complex genes including the light harvesting complex were differentially expressed in 'HR' in both GO and KEGG analysis, while in 'HS', genes involved in photosynthesis were identified only in KEGG analysis. These results may contribute to explaining the reasons why 'HR' is more tolerant than 'HS' under heat treatment, and exploring the mechanisms of heat resistance.

It is well known that HSP family exert crucial roles in responding to heat stress and could be immediately induced by high temperatures [48–50]. The HSPs can be divided into five structurally distinct classes, including: HSP100, HSP90, HSP70, HSP60, and small HSPs (sHSPs) [51]. Small HSPs, which are especially abundant in plants, plays important roles in abiotic stress tolerance and are found to be in great abundance and diversity [52]. Overexpression of a small heat shock protein, CaHSP16.4, of pepper (*Capsicum annuum* L.) enhances tolerance to heat stress in *Arabidopsis thaliana* [53]. Wang et al. [54] reported that the expression of protein HSP21 was induced by heat stress in grapevine leaves, and decreased in the control during the recovery. Overexpression of EaHSP70 gene from *Erianthus arundinaceus* confers drought and salinity tolerance in sugarcane [55]. In this study, most HSP genes including HSP90s, HSP70s and sHSPs were up-regulated in jujube after heat stress (Fig 6). Other studies have reported that HSPs are critical for plants to acclimate to heat stress [56]. Thus, the up-regulated HSP genes play important roles in heat tolerance in jujube.

Many transcription factors (TF) have been reported to act as important factors in regulating the expression of specific downstream genes in abiotic stresses, such as high temperatures, cold, high salinity, and drought [57,58]. Therefore, identification and characterization of TFs involved in abiotic stress response is crucial to reveal the molecular mechanisms. In this study, we also identified some differentially expressed transcription factors in response to heat stress, including ERF, WRKY, MYB, NAC, DERB, bHLH, HSF and C2H2 (Table 2). The highest number of up-regulated genes between the 'HR' and 'HS' were ERF transcription factors, which was consistent with the results on Korean fir (*Abies koreana*) under heat stress [56]. Overexpression of *AhERF019* gene from peanuts (*Arachis hypogaea* L.) conferred tolerance to heat stress in *Arabidopsis* [59]. Plant-specific NAC transcription factors have been reported to act as key regulators in abiotic stress responses in recent years [60]. Overexpressing *TaNAC2L* from wheat (*Triticum aestivum*) improved acquired heat tolerance and activated the expression of heat-related genes in transgenic *Arabidopsis* plants [61]. In our study, five NAC TFs (LOC107419222, LOC107421097, LOC107428947, LOC107435293, LOC107406551) and two NAC TFs (LOC107406551, LOC107416163) were up-regulated in 'HR' and 'HS' for all the heat treatments (S6 Table), respectively, and NAC72 (LOC107406551) was higher expressed in 'HR' and 'HS' after heat stress (S3 Table). In addition, the NAC2 (LOC107421097) gene was highly expressed in the heat-resistant jujube cultivar 'HR' at all the time points of heat stress. These results suggested that NACs participated in the heat stress response and played different roles in the expression patterns between the heat-resistant and heat-sensitive cultivars.

HSFs have been shown to be involved in response to heat stress [62]. Most plant HSFs are mostly regulated by heat stress. In pepper (*Capsicum annum* L.), HSFA3 was more induced by heat stress in the heat-susceptible cultivar 'S590', and HSFA2 was upregulated in the heat-tolerant cultivar 'R597' [22]. Yan et al. [21] reported that fourteen HSFs were induced in spinach leaves during the heat treatment. In our study, we found one heat stress transcription factor C1 (LOC107431837) was up-regulated in both 'HR' and 'HS' at all the time points of heat stress, and the expression of *HSFC1* (LOC107431837) showed an dramatic increase in 'HR' especially after 3d, 5d of heat stress (S3 Table), which indicated that the high expression of *HSFC1* in 'HR' might contribute to its tolerance to heat stress to some extent.

The ubiquitin-proteasome system (UPS) is a central regulator that controls plant response and adaptation to environmental stresses [63]. The UPS is composed of three enzymes: E1 (ubiquitin activating enzymes), E2 (ubiquitin conjugating enzymes), and E3 (ubiquitin ligases) [64]. Overexpression of a rice RING finger E3 ligase, OsHCI1, confers enhanced heat tolerance in *Arabidopsis* [65]. Morimoto et al. [66] found that BPM-CUL3 E3 ligase could modulate the heat stress response and prevent an adverse effect of excess DREB2A on plant growth in *Arabidopsis*. The expression of six ubiquitin-protein ligase genes were detected to be significantly increased in the heat-tolerant chieh-qua cultivar 'A39' when compared with the heat-sensitive cultivar 'H5' after four days of heat stress [50]. In the present study, we identified nineteen ubiquitin-protein ligase genes, among which, the expressions of fifteen genes (LOC107428626, LOC107424969, LOC107413314, LOC107413313, LOC107422277, LOC107426882, LOC107420280, LOC107415107, LOC107432321, LOC107419944, LOC107404052, LOC107431904, LOC107411111, LOC107431883, LOC107403709)were all higher in 'HR' than in 'HS' at 3, 5d (Fig 7, S3 Table), indicating that the high expression of these ubiquitin-protein ligases after prolonged heat stress may help 'HR' confer stronger adaptability to high temperature than 'HS'.

## Conclusions

In this study, transcriptome analysis was performed on heat-resistant cultivar 'HR' and heat-sensitive cultivar 'HS' after 0,1, 3, and 5d of heat stress. Totally, 1880, 4350, 5435 differentially expressed genes were identified in 'HR', and 2390, 2998, 4064 genes were detected in 'HS' after heat stress for 1, 3 and 5d, respectively. Some DEGs related to heat shock proteins, transcription factors, and ubiquitin-protein ligase genes were identified during heat stress. Therefore, this study not only provided a basis for further understanding of the molecular mechanism on heat tolerance of jujube plants, but also exerted valuable and useful genes involved in heat stress, which would be helpful for the genetic improvement of heat tolerance in jujube breeding.

## Supporting information

**S1 Table. Primers used in the study for quantitative RT-PCR.**
(XLS)

**S2 Table. Summary of the sequencing results.**
(XLS)

**S3 Table. List of DEGs in 'HR' and 'HS'.**
(XLSX)

**S4 Table. List of DEGs between 'HR' and 'HS' in different comparisons.**
(XLSX)

**S5 Table. GO enrichment of DEGs in 'HR' and 'HS'.**
(XLSX)

**S6 Table. Pathway analysis of DEGs in 'HR' and 'HS'.**
(XLSX)

**S7 Table. Transcription factors involved in heat stress.**
(XLSX)

## Acknowledgments

We would like to thank Biomarker Technologies Co., Ltd., Beijing, China for technical help.

## Author Contributions

**Conceptualization:** Juan Jin, Xuxin Liu, Qing Hao.

**Data curation:** Juan Jin, Lei Yang, Dingyu Fan.

**Formal analysis:** Juan Jin.

**Funding acquisition:** Juan Jin.

**Methodology:** Lei Yang, Dingyu Fan.

**Supervision:** Qing Hao.

**Writing – original draft:** Juan Jin.

**Writing – review & editing:** Juan Jin.

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
