## [Decision Letter · Decision Letter 0]

25 Feb 2020

PONE-D-20-01047

Comparative Transcriptome Analysis Uncovers Different Heat Stress Responses in Heat-resistant and Heat-sensitive Jujube Cultivars

PLOS ONE

Dear Mrs. juan,

Thank you for submitting your manuscript to PLOS ONE. After careful consideration, we feel that it has merit but does not fully meet PLOS ONE’s publication criteria as it currently stands. Therefore, we invite you to submit a revised version of the manuscript that addresses the points raised during the review process.

We would appreciate receiving your revised manuscript by Apr 10 2020 11:59PM. To enhance the reproducibility of your results, we recommend that if applicable you deposit your laboratory protocols in protocols.io, where a protocol can be assigned its own identifier (DOI) such that it can be cited independently in the future. For instructions see: http://journals.plos.org/plosone/s/submission-guidelines#loc-laboratory-protocols

We look forward to receiving your revised manuscript.

Kind regards,

Eric A Shelden, Ph.D.

Academic Editor

PLOS ONE

Journal Requirements:

Reviewers' comments:

Reviewer's Responses to Questions

**Comments to the Author**

1. Is the manuscript technically sound, and do the data support the conclusions?

Reviewer #1: Partly

Reviewer #2: No

2. Has the statistical analysis been performed appropriately and rigorously? 

Reviewer #1: Yes

Reviewer #2: Yes

3. Have the authors made all data underlying the findings in their manuscript fully available?

Reviewer #1: Yes

Reviewer #2: Yes

4. Is the manuscript presented in an intelligible fashion and written in standard English?

Reviewer #1: Yes

Reviewer #2: Yes

5. Review Comments to the Author

Reviewer #1: The manuscript makes a contribution to understanding genes underlying heat stress resistance in jujube. The authors have done a good job with the RNA-Seq analysis, however, additional analysis and interpretation of the data is needed. The manuscript PDF has been marked with reviewer comments and is attached.

Reviewer #2: In this manuscript the authors compare gene expression analysis between two Jujube cultivars under heat stress. The results of gene differential expression are coupled to a few additional observations for confirming the occurrence of heat stress during treatments at 45°C.

The manuscript lacks key information:

1. How did the authors select ‘HR’ and ‘HS’ Ziziphus jujube cultivars?

2. How did they set the heat treatment? Why did they choose 45 °C for heat treatments?

3. How many plants did they heat-stress and from how many plants did they collect the material? In the results section they wrote that they performed three biological replicate, however it is not clear what they mean with biological replicate.They also wrote that the validation of DEGs was done in the same plant samples, but again it is not clear on how many plants.

4. It is also unclear on how many plants they did the TEM observations.

5. In results the authors report" the nucleoli of two jujube cultivars gradually disappeared as the heat

stress time extended, but the nuclear membranes remained intact...." What does this mean biologically. In this chapter the discussion is missing and it is scarse also in the two chapters describing DE transcription factors and HSP70. In my opinion a discussion on the functional outputs of DEGs must be added in the manuscript that tends to be too superficial.

6. PLOS authors have the option to publish the peer review history of their article (what does this mean?). If published, this will include your full peer review and any attached files.

Reviewer #1: No

Reviewer #2: No

---

## [Author Response · Author response to Decision Letter 0]

1 Apr 2020

Dear editors,

The authors are very thankful to you and the reviewers for the valuable comments and remarks regarding this manuscript. We have tried to address all comments and suggestions adequately. The requested alterations/corrections have been inserted directly into the manuscript and are also described below.

We are looking forward to having the opportunity to publish our work on your journal. Thank you for your help again!

Best regards!

Comments from Reviewers:

Reviewer #1: The manuscript makes a contribution to understanding genes underlying heat stress resistance in jujube. The authors have done a good job with the RNA-Seq analysis, however, additional analysis and interpretation of the data is needed. The manuscript PDF has been marked with reviewer comments and is attached.

1. Change to "genes are involved".

Response: "genes involved" was changed to "genes are involved" in the revised manuscript.

2. Change capital "S" to lower case "s".

Response: capital "S" was changed to lower case "s" in the revised manuscript.

3. Provide more information on "HR" and "HS". Are these different jujube cultivars?

Response: Ziziphus jujuba cv.‘HR’ and Ziziphus jujuba cv.‘HS’ were grown in the Forestry Management Station of Turpan city, Forestry and Grassland Administration of Turpan city, located in Turpan, Xinjiang province, China. ‘HR’ has been proven to be resistant to heat stress while ‘HS’ is susceptible.

4. Indicate if these are three replicate plant for each of the time points of the heat treatment. This would then be a total of 12 plants for each of HR and HS.

Response: There were a total of 180 plants for ‘HR’ and ‘HS’ seedlings, respectively. Each treatment included three biological replications for each cultivar, and each biological replicate contained 15 plants. Plant samples were collected after the treatments started.

5. Indicate if the physiological indexes were measured for replicate plants in each treatment time point and control.

Response: Three physiological indexes, including electrolyte leakage, malondialdehyde (MDA), and proline were measured 3 times (n=3) with 5 plants per replicate in each treatment time point for each cultivar in order to analyze the physiological changes of two jujube cultivars subjected to heat stress.

6. Change to "Construction of RNA."

Response: "RNA" was changed to " Construction of RNA " in the revised manuscript.

7. How do the physicochemical changes in jujube compare to other plants? 

Response: Our ultrastructural observation clearly showed that the time-course heat stress exerted a more conspicuous effect on the heat-sensitive jujube cultivar ‘HS’ than heat-resistant jujube cultivar ‘HR’, the chloroplasts became more globular and the number of osmiophilic granules increased in response to heat, which were consistent with the leaf ultrastructure changes of Ara et al. 2015 in cucurbits. Moreover, we found that the starch grains of ‘HR’ were faintly visible, while the number of starch granules in ‘HS’ decreased and gradually disappeared, this phenomenon may be correlated with glucose starvation upon a decrease in photosynthetic activity under heat stress (Jin et al., 2011), confirming that ‘HS’ was susceptible to heat stress. The physiological indexes showed that heat stress increased the relative electrolyte leakage and MDA content in the two jujube cultivars, however, ‘HS’ suffered a greater degree of membrane damage than ‘HR’ (Fig.1). Similar results were reported by Kumar et al., 2012 and Li et al. (2015). Under heat stress conditions, the accumulation of proline may act as an osmoprotectant for cellular structures in response to high temperature, which might contribute to greater heat tolerance (Kumar et al., 2012). Our results showed that higher proline content was accumulated in heat-tolerant jujube cultivar ‘HR’, this finding confirmed the results of study on the heat tolerant genotypes, such as maize (Kumar et al., 2012) and sorghum (Gosavi et al.,2014), indicating that proline can be strongly correlated with the capacity of genotypes to survive heat stress. These discussions were added in the revised manuscript.

8. The text marking cellular structures on the figure was too small to read.

Response: We have changed the text font marking cellular structures on the figure in the revised manuscript.

9. Change to "RNASeq".

Response: We have changed "RNA" to "RNASeq" in the revised manuscript.

10. What were these five genes?

Response: These five genes were LOC107406551, LOC107434446, LOC107412389, Ziziphus_jujuba_newGene_4043, Ziziphus_jujuba_newGene_7074. These results were added in the revised manuscript.

11. The GO analysis is presented as bar graphs of DEG and All genes. The authors should use a statistical determination of the significance of GO term enrichment in for DEG. Software such as topGO implemented in R can provide such an analysis.

Response: We have used a statistical determination of the significance of GO term enrichment in for DEG and be presented as a graph (Figure 5) in the revised manuscript.

12. It is recommended that the KEGG analysis be presented as a bar graph or table of KEGG terms and their corrected p-values for those terms with corrected p-values less than 0.05.

Response: We have presented the KEGG analysis as a bar graph (Figure 6) in the revised manuscript.

13. A figure (like Fig. 6) showing the data on the HSP genes in the main text is recommended.

Response: We have changed the table to a figure showing the data on the HSP genes in the revised manuscript.

14. The RNA-seq data should be presented in a bar graph similarly to the qRT-PCR. Log transformation of the data is recommended.

Response: The RNA-seq data has been changed to a bar graph similarly to the qRT-PCR in the revised manuscript. 

15. Why were the HSPs, TF and Ub-ligases of such interest to the authors. How may their expression contribute to heat stress resistance? What is the molecular function of these different classes of genes and how does this function contribute to resistance to heat stress? Are the heat stress responses in HR and HS jujube plants similar to other plants? What was the most important biological process or pathway identified in the study as associated with heat susceptibility or tolerance? What is the most important difference found between heat susceptible and heat resistant cultivars?

Response: According to the previous research results, analysis of the transcriptome profile in plant after heat treatment indicated that the HSP family and transcription factors (TFs) play a central role in responding to heat stress. Moreover, researchers have reported that Ub-ligases genes confers enhanced heat tolerance, so we’re interested in these genes, and we have also added relevant content in the discussion section in the revised manuscript.

Reviewer #2: In this manuscript the authors compare gene expression analysis between two jujube cultivars under heat stress. The results of gene differential expression are coupled to a few additional observations for confirming the occurrence of heat stress during treatments at 45°C.

1. How did the authors select ‘HR’ and ‘HS’ Ziziphus jujube cultivars?

Response: Ziziphus jujuba cv.‘HR’ and Ziziphus jujuba cv.‘HS’ were grown in the Forestry Management Station of Turpan city, Forestry and Grassland Administration of Turpan city, located in Turpan, Xinjiang province, China. ‘HR’ has been proven to be resistant to heat stress while ‘HS’ is susceptible.

2. How did they set the heat treatment? Why did they choose 45 °C for heat treatments?

Response: In recent years, with the global climate warming, extreme high temperature weather frequently occurs in Xinjiang, China, and many areas have experienced continuous high temperature weather exceeding 40 ℃, which has greatly affected the growth and yield of jujube trees. In the early stage, we have conducted an evaluation test on the heat resistance of jujube leaves, and determined the treatment temperature (45 °C).

3. How many plants did they heat-stress and from how many plants did they collect the material? In the results section they wrote that they performed three biological replicate, however it is not clear what they mean with biological replicate. They also wrote that the validation of DEGs was done in the same plant samples, but again it is not clear on how many plants.

Response: There were a total of 180 plants for ‘HR’ and ‘HS’ seedlings, respectively. Each treatment included three biological replications for each cultivar, and each biological replicate contained 15 plants. Plant samples were collected after the treatments started.

4. It is also unclear on how many plants they did the TEM observations.

Response: For ultra-structural observation, the sixth leaf from the top was removed from 4 random plants in each treatment time point (n=3) for each cultivar.

5. In results the authors report" the nucleoli of two jujube cultivars gradually disappeared as the heat

stress time extended, but the nuclear membranes remained intact...." What does this mean biologically. In this chapter the discussion is missing and it is scarse also in the two chapters describing transcription factors and HSP70. In my opinion a discussion on the functional outputs of DEGs must be added in the manuscript that tends to be too superficial.

Response: We have revised the discussion section in the revised manuscript, including physiochemical changes, transcription factors, HSP70 and ubiquitin-protein ligase genes.

---

## [Decision Letter · Decision Letter 1]

24 Apr 2020

PONE-D-20-01047R1

Comparative Transcriptome Analysis Uncovers Different Heat Stress Responses in Heat-resistant and Heat-sensitive Jujube Cultivars

PLOS ONE

Dear Mrs. juan,

Thank you for submitting your manuscript to PLOS ONE. After careful consideration, we feel that it has merit and has been significantly improved but does not fully meet PLOS ONE’s publication criteria as it currently stands. Therefore, we invite you to submit a revised version of the manuscript that addresses the remaining points raised during the review process.

We would appreciate receiving your revised manuscript by Jun 08 2020 11:59PM. To enhance the reproducibility of your results, we recommend that if applicable you deposit your laboratory protocols in protocols.io, where a protocol can be assigned its own identifier (DOI) such that it can be cited independently in the future. For instructions see: http://journals.plos.org/plosone/s/submission-guidelines#loc-laboratory-protocols

We look forward to receiving your revised manuscript.

Kind regards,

Eric A Shelden, Ph.D.

Academic Editor

PLOS ONE

Reviewers' comments:

Reviewer's Responses to Questions

**Comments to the Author**

1. If the authors have adequately addressed your comments raised in a previous round of review and you feel that this manuscript is now acceptable for publication, you may indicate that here to bypass the “Comments to the Author” section, enter your conflict of interest statement in the “Confidential to Editor” section, and submit your "Accept" recommendation.

Reviewer #1: (No Response)

Reviewer #2: (No Response)

2. Is the manuscript technically sound, and do the data support the conclusions?

Reviewer #1: Yes

Reviewer #2: Yes

3. Has the statistical analysis been performed appropriately and rigorously? 

Reviewer #1: Yes

Reviewer #2: Yes

4. Have the authors made all data underlying the findings in their manuscript fully available?

Reviewer #1: Yes

Reviewer #2: Yes

5. Is the manuscript presented in an intelligible fashion and written in standard English?

Reviewer #1: Yes

Reviewer #2: Yes

6. Review Comments to the Author

Reviewer #1: The manuscript is much improved particularly the Discussion section. There are some further revisions recommended for this version.

Abstract – . A sentence on how stress was quantified in the study should be added. The section starting with “in total, 6887… 4064 genes were significantly different in ‘HR’ and ‘HS’, respectively.” should be replaced with sentences indicating the functions of the differentially expressed genes at each time point that were discovered in the GO and KEGG analyses.

The text is difficult to read in Figs 2-6.

Figure 5, 6 and 9 Captions: Provide a description of what the axes are on these graphs.

Indicate what the columns of the supplementary files are in the caption included at the top of the excel spreadsheets.

Some additional discussion of GO and KEGG analyses is recommended. For example, photosynthetic complex genes including the light harvesting complex are differentially expressed in HR, but HS in both GO and KEGG analysis.

Line spacing is uneven in the document.

Reviewer #2: The authors did not respond to my previous observation no. 5. In results the authors report" the nucleoli of two jujube cultivars gradually disappeared as the heat stress time extended, but the nuclear membranes remained intact".What does this mean biologically?

I think the authors should discuss this point and consider the literature available on nucleoli and stress.

7. PLOS authors have the option to publish the peer review history of their article (what does this mean?). If published, this will include your full peer review and any attached files.

Reviewer #1: No

Reviewer #2: No

---

## [Author Response · Author response to Decision Letter 1]

20 May 2020

Dear editors,

The authors are very thankful to you and the reviewers for the valuable comments and remarks regarding this manuscript. We have tried to address all comments and suggestions adequately. The requested alterations/corrections have been inserted directly into the manuscript and are also described below.

We are looking forward to having the opportunity to publish our work on your journal. Thank you for your help again!

Best regards!

Comments from Reviewers:

Reviewer #1: The manuscript is much improved particularly the Discussion section. There are some further revisions recommended for this version.

1. Abstract: A sentence on how stress was quantified in the study should be added. The section starting with “in total, 6887… 4064 genes were significantly different in ‘HR’ and ‘HS’, respectively.” should be replaced with sentences indicating the functions of the differentially expressed genes at each time point that were discovered in the GO and KEGG analyses.

Response: “In total, 6887 and 5077 differentially expressed genes were identified in ‘HR’ and ‘HS’, respectively. After 1d of heat stress, 1880 and 2390 genes presented significantly different expression in ‘HR’ and ‘HS’ compared with the control; after 3d of heat stress, 4350 and 2998 genes were differentially expressed in ‘HR’ and ‘HS’; after 5d of heat stress, expression of 5435 and 4064 genes were significantly different in ‘HR’ and ‘HS’, respectively. The expression pattern of nine genes was validated by qRT-PCR. After gene ontology and KEGG enrichment analysis, the DEGs encoding heat shock proteins, transcriptional factors, and ubiquitin-protein ligase genes were found to be closely involved in heat stress response” was changed to “A total of 6887 and 5077 differentially expressed genes were identified in ‘HR’ and ‘HS’ after 1d, 3d, and 5d of heat stress compared with the control treatment, GO and KEGG enrichment analysis revealed that some of the genes were highly enriched in oxidation-reduction process, response to stress, response to water deprivation, response to heat, carbon metabolism, protein processing in endoplasmic reticulum, and plant hormone signal transduction and may play vital roles in the heat stress response in jujube plants. Differentially expressed genes were identified in the two cultivars, including heat shock proteins, transcriptional factors, and ubiquitin-protein ligase genes. And the expression pattern of nine genes was also validated by qRT-PCR”.

2. The text is difficult to read in Figs 2-6.

Response: We have changed the text font on Figs 2-6 in the revised manuscript.

3. Figure 5, 6 and 9 Captions: Provide a description of what the axes are on these graphs.

Response: We have revised the axes on Figure 5, 6 and 9 in the revised manuscript. 

Figure 5. Scatterplot of enriched GO pathways between the two cultivars after heat stress. (A), (C) and (E) represent enriched GO pathways of the DEGs in HR 1d vs. HR0d, HR 3d vs. HR0d, and HR 5d vs. HR0d, respectively; (B), (D) and (E) represent enriched GO pathways of the DEGs in HS 1d vs. HS0d, HS 3d vs. HS0d, and HS 5d vs. HS0d, respectively. The horizontal axes represent the enriched GO pathways, vertical axes represent the GeneRatio of each GO pathway. GeneRatio refers to the ratio of the number of DEGs enriched in certain GO pathway to the total number of differentially expressed genes. The greater the value is, the higher the number of DEGs is. The size of the dots indicates the number of DEGs enriched in certain pathway, and the color of the dots corresponds to the range of the q value (adjusted p value). Only the top 20 terms are listed here.

Figure 6. Scatterplot of enriched KEGG pathways between the two cultivars after heat stress. (A), (C) and (E) represent enriched KEGG pathways of the DEGs in HR 1d vs. HR0d, HR 3d vs. HR0d, and HR 5d vs. HR0d, respectively; (B), (D) and (E) represent enriched KEGG pathways of the DEGs in HS 1d vs. HS0d, HS 3d vs. HS0d, and HS 5d vs. HS0d, respectively. The horizontal axes represent the enriched KEGG pathways, vertical axes represent the number of DEGs enriched in each KEGG pathway. The bar indicates the number of DEGs enriched in certain pathway, and the color of the bars corresponds to the range of the q value (adjusted p value). The longer the bar is, the more the number of DEGs is. Only the top 20 terms are listed here.

Figure 9. Expression analysis of nine selected DEGs in ‘HR’ and ‘HS’ after 0, 1, 3, 5d of heat stress. (A) qRT-PCR results of nine DEGs. (B) mRNA expression levels of the same DEGs determined by RNA-seq, Log2(FPKM) was used to calculate the expression levels of genes. Horizontal axes represent 0, 1, 3, and 5 d (from left to right) of heat stress in HR and HS, the vertical axes represent the expression levels. The white bar presents the expression levels in ‘HR’, and the grey bar presents the expression levels in ‘HS’. The error bars represent SD (n=3).

4. Indicate what the columns of the supplementary files are in the caption included at the top of the excel spreadsheets.

Response: We have renamed the columns of the supplementary files clearly and added the captions at the top of the excel spreadsheets in the revised manuscript.

5.Some additional discussion of GO and KEGG analyses is recommended. For example, photosynthetic complex genes including the light harvesting complex are differentially expressed in HR, but HS in both GO and KEGG analysis.

Response: We have added some discussion of GO and KEGG analyses in the revised manuscript.

6. Line spacing is uneven in the document.

Response: We have changed the line spacing to the double-space paragraph format in the revised manuscript.

Reviewer #2: The authors did not respond to my previous observation no. 5. In results the authors report" the nucleoli of two jujube cultivars gradually disappeared as the heat stress time extended, but the nuclear membranes remained intact". What does this mean biologically?

I think the authors should discuss this point and consider the literature available on nucleoli and stress.

Response: We have discussed this point in the revised manuscript. The nucleolus has been considered as a sensor of cell stress since it can suffer alterations of morphology, area, and number per nucleus in response to stress (Boulon et al. 2010; Jiang et al. 2014; Stepinski. 2014). In our study, we found that the nucleoli of two jujube cultivars were sensitive to heat stress and gradually disappeared, which had been reported in cowpea cells after exposure to elevated temperature of 45 ℃ (Dylewski et al., 1991).

---

## [Decision Letter · Decision Letter 2]

23 Jun 2020

Comparative Transcriptome Analysis Uncovers Different Heat Stress Responses in Heat-resistant and Heat-sensitive Jujube Cultivars

PONE-D-20-01047R2

Dear Dr. hao,

We’re pleased to inform you that your manuscript has been judged scientifically suitable for publication and will be formally accepted for publication once it meets all outstanding technical requirements.

Kind regards,

Eric A Shelden, Ph.D.

Academic Editor

PLOS ONE

Additional Editor Comments (optional):

Reviewers' comments:

Reviewer's Responses to Questions

**Comments to the Author**

1. If the authors have adequately addressed your comments raised in a previous round of review and you feel that this manuscript is now acceptable for publication, you may indicate that here to bypass the “Comments to the Author” section, enter your conflict of interest statement in the “Confidential to Editor” section, and submit your "Accept" recommendation.

Reviewer #1: All comments have been addressed

Reviewer #2: All comments have been addressed

2. Is the manuscript technically sound, and do the data support the conclusions?

Reviewer #1: Yes

Reviewer #2: (No Response)

3. Has the statistical analysis been performed appropriately and rigorously? 

Reviewer #1: Yes

Reviewer #2: (No Response)

4. Have the authors made all data underlying the findings in their manuscript fully available?

Reviewer #1: Yes

Reviewer #2: (No Response)

5. Is the manuscript presented in an intelligible fashion and written in standard English?

Reviewer #1: Yes

Reviewer #2: (No Response)

6. Review Comments to the Author

Reviewer #1: Authors have addressed concerns in previous reviews. There remains a few typos in the text and a thorough copy editing is needed.

Reviewer #2: (No Response)

7. PLOS authors have the option to publish the peer review history of their article (what does this mean?). If published, this will include your full peer review and any attached files.

Reviewer #1: No

Reviewer #2: No

---

## [Editor Report · Acceptance letter]

8 Jul 2020

PONE-D-20-01047R2 

Comparative Transcriptome Analysis Uncovers Different Heat Stress Responses in Heat-resistant and Heat-sensitive Jujube Cultivars 

Dear Dr. Hao:

I'm pleased to inform you that your manuscript has been deemed suitable for publication in PLOS ONE. Congratulations! Your manuscript is now with our production department. 

Kind regards, 

on behalf of

Dr. Eric A Shelden 

Academic Editor

PLOS ONE